# Assessing Biotic and Abiotic Effects on Biodiversity Index Using Machine Learning

Mahmoud Bayat [1,*], Harold Burkhart [2], Manouchehr Namiranian [3], Seyedeh Kosar Hamidi [4], Sahar Heidari [5] and Majid Hassani [1]

1 Research Institute of Forests and Rangelands, Agricultural Research, Education and Extension Organization (AREEO), Tehran 14968-13111, Iran; Hassani@gmail.com

2 Department of Forest Resources and Environmental Conservation, Virginia Polytechnic Institute and State University, 319 Cheatham Hall, 310 West Campus Drive, Blacksburg, VA 24061, USA; burkhart@vt.edu

3 Department of Forestry, Faculty of Natural Resources, University of Tehran, Tehran 77871-31587, Iran; Mnamiri@ut.ac.ir

4 Department of Forestry, Faculty of Natural Resources, Sari Agriculture Sciences and Natural Resource University, Sari 48181-66996, Iran; k.hamidi86@yahoo.com

5 Department of Environment, Faculty of Natural Resources, University of Tehran, Tehran 77871-31587, Iran; saharheidari@ut.ac.ir

* Correspondence: Mbayat@rifr-ac.ir

**Abstract:** Forest ecosystems play multiple important roles in meeting the habitat needs of different organisms and providing a variety of services to humans. Biodiversity is one of the structural features in dynamic and complex forest ecosystems. One of the most challenging issues in assessing forest ecosystems is understanding the relationship between biodiversity and environmental factors. The aim of this study was to investigate the effect of biotic and abiotic factors on tree diversity of Hyrcanian forests in northern Iran. For this purpose, we analyzed tree diversity in 8 forest sites in different locations from east to west of the Caspian Sea. 15,988 trees were measured in 655 circular permanent sample plots (0.1 ha). A combination of machine learning methods was used for modeling and investigating the relationship between tree diversity and biotic and abiotic factors. Machine learning models included generalized additive models (GAMs), support vector machine (SVM), random forest (RF) and K-nearest–neighbor (KNN). To determine the most important factors related to tree diversity we used from variables such as the average diameter at breast height (DBH) in the plot, basal area in largest trees (BAL), basal area (BA), number of trees per hectare, tree species, slope, aspect and elevation. A comparison of RMSEs, relative RMSEs, and the coefficients of determination of the different methods, showed that the random forest (RF) method resulted in the best models among all those tested. Based on the results of the RF method, elevation, BA and BAL were recognized as the most influential factors defining variation of tree diversity.

**Keywords:** elevation; aspect; slope; modeling tree diversity; random forest

## 1. Introduction

Forests are one of the most important terrestrial ecosystems, providing a variety of services to humans [1]. Forests play multiple important roles in meeting the habitat needs of different organisms [2]. Biodiversity is a structural feature in dynamic and complex forest ecosystems. [3]. One of the most challenging issues in modeling forest ecosystems is understanding the relationship between biodiversity and environmental factors. Although this has been the focus of many investigations in recent years and extensive research has been done in this field [4,5], much work remains to be done. Biodiversity analysis in a forest along the gradient of environmental factors can provide meaningful information about the structure of the forest community. For example, an analysis of altitudinal gradient concluded that functional divergence shows a single peak change law with increasing

elevation in the Baihua Mountains [6]. Species diversity generally decreases with increasing latitude [5]. Factors affecting species diversity are not only abiotic and environmental, but also include environmental gradients by biotic factors such as competition [7]. Biotic conditions affect species composition and create differences in the diversity of the community [8]. Biotic factors related to the structure of a community have a significant effect on access to niche in coexisting species [9]. Composition of forest ecosystems is the result of biotic interactions coupled with environmental influences [5].

Many studies have considered abiotic factors of forest by numerical methods and related them to tree growth measured in the field [10–14]. Furthermore, various studies have considered only biotic factors and their effect on forest biodiversity [15,16]. In these cases, abiotic environmental effects on species distributions and their ability to sustain viable populations in specific environmental configurations are not evaluated [17]. However, the species diversity in a given site is also influenced by other species through interactions such as trophic, non-trophic and competitive [18,19].

Few studies have considered the impact of biotic and abiotic factors on biodiversity that change simultaneously [20,21]. Some studies indicate that biodiversity of a forest is affected by many factors, such as local climatic conditions, soil characteristics, biodiversity, and even the type of management practices employed [6,22]. Environmental factors are key variables that can help determine the diversity and distribution of plant species. For example, an analysis of patterns of diversity across different climate conditions of forest in China, which used PCA analysis to build the compound habitat gradient and biotic and abiotic factors, concluded that both biotic and abiotic factors influence diversity of the forest community across different climatic zones [23].

Research studies that have considered influence of both biotic and abiotic factors on biodiversity are limited, and this is one of the unique aspects of this study.

In recent years, machine learning (ML) methods have been widely used to investigate the effect of changes in environmental conditions on biodiversity. ML is one of the most important branches of artificial intelligence, which is widely used for analyzing forest data due to its significant advantages in some circumstances. ML processes often include the following: (i) Data selection and preprocessing. (ii) algorithm selection and (iii) assessment of solutions [24,25]. The application of machine learning has been used to improve local, regional, and global estimates in complex ecosystems such as forests [26]. ML methods have also been used in forest management and hazard assessment [27,28].

ML models can provide good accuracy and capacity in solving complex issues and representing nonlinear behavior of systems; these methods require rigorous training and test data [29]. However, long-term, accurate monitoring can be costly, and data collection, storage, and updating can be disrupted. Therefore, researchers have proposed different ML methods as well as a combination of ML methods and traditional methods to better understand the different environmental mechanisms in forests as well as to solve different problems [27,30]. It is important to note that there is no one universal suitable ML method for all studies, and the choice of the most appropriate and best method, or a combination of the methods, depends on the users and their objectives [31]. For example survival and height of trees were predicted and modeled with ML, respectively, and determined the affecting factors in natural forests [32,33].

In complex ecosystems such as forests, ML methods have been used extensively to improve global, regional, and local forecasts. Machine learning and geo statistical methods were used to predict aboveground biomass (AGB) in Chinese forests [34]. They concluded that the proposed random forest approach provided a reliable and accurate method for AGB mapping in subtropical forest regions with complex topography.

ML models have many advantages that justify their choice for modeling forest features such as biodiversity. For example, the random forest approach is less sensitive to parameter adjustments than other ML models, and it provides an assessment of the importance of variables. This model is more powerful in reducing data than other models, and it is

more accurate than decision trees; it also generates an internal unbiased estimate of the generalized error as model building progresses [35,36].

The SVM model is another ML model that was used in this research. SVM models can be applied for solving nonlinear, regression and density estimation problems, and they are very useful in forest modeling. In addition, they use kernel functions in the form of points to project the multidimensional space of data and then find the best classification of the hyperplane [37].

The K-nearest neighbors (KNN) model was also applied in this study. KNN is one of the easiest and simplest ML models, which is one of the advantages of using it. In this model, no hypothesis about the distribution of prediction variables is required, and it can be applied to both single and multivariate predictions [38].

Another method, the GAM model, was also considered as it can limit the error in prediction of a dependent variable Y by assessing unspecified functions, which are connected by means of a link function with the dependent variable. By defining the model in terms of a smooth function the GAM model, provides a flexible specification of response [39].

It can be said that the key to reveal mechanisms for conserving species diversity and formulating forest management strategies is to study the response of species diversity to changes in environmental conditions and its relationship with ecosystem performance [40].

Hyrcanian forests in the form of green strips are green belt forests that are often temperate. These forests are located along the southern borders of the Caspian Sea and over the northern slopes of the Alborz Mountain. These forests extend from Astara in northwestern Iran to Golidaghi in Gorgan in northeastern Iran and are located in the three provinces of Gilan, Mazandaran and Golestan. According to the latest statistics of the Forests and Rangelands Organization of Iran (FRWO), these forests cover an area of 1.85 million hectares, which constitute 1.1% of the country and 15% of the total forests of Iran. Hyrcanian forests, with more than 3200 plant species, have significant biodiversity on a global level. This region is of special importance with about 44% of vascular plants in Iran. There are about 500 species of plants native to Iran in these forests. All these features emphasize the need to protect the biodiversity of this region [41]. The main threat to biodiversity of the Hyrcanian forest arises from habitat loss and fragmentation resulting from conversion of forests to urban development, agricultural land, dam construction, private dwellings, and other non-forest uses.

The aim of this study was to investigate the effect of biotic and abiotic factors on the tree diversity of Hyrcanian forests in northern Iran. For this purpose, a combination of ML models was used. ML models include GAM, SVM, RF and KNN and try to include the most important factors such as average DBH in plot, BAL, BA, number of tree per ha, tree species and slope, aspect and elevation. This study is unique in that the selected forest sites are comprehensive and cover the Hyrcanina forest from west to east. Past studies have only included a limited part of these forests in terms of biodiversity, this study is the first to completely cover these forests by selecting 8 forest sites from east to west. In addition to diversity modeling, we analyzed tree diversity differences in the east-west gradient of Hyrcanian forests.

## 2. Materials and Methods

### 2.1. Study Area

The forests of northern Iran, sometimes referred to as ancient forests, cover an area of about 1,848,000 ha on the southern border of the Caspian Sea and along the northern slopes of the Alborz mountain range from Astara in the west to Golidaghi in the east. The main tree species in the Hyrcanian forests are *Fagus orientalis* Lipsky, *Carpinus betulus* L., *Tilia platyphyllos* Scop., *Acer velutinum* Boiss, *Alnus subcordata* C.A. Mey, *Quercus castaeifolia*, *Fraxinus excelsior* L., *Cerasus avium* (L.) Moench, *Sorbus terminalis* (L.) Grantz, *Ulmus glabra* Hudson, *Acer cappadicium* Gled, *Parrotia persica* C.A. Mey, *Diospyros lotus*, *Ulmus minor* Miller, *Petrocarya fraxinifolia* (Lam.), and *Taxus baccata*. Regeneration of oriental beech

(*Fagus orientalis*), a preferred tree species, seems to be occurring sufficiently, even in light of competition from European hornbeam (*Carpinus betulus*) [42].

In terms of global division, these forests are deciduous forests with a humid sub-Mediterranean climate [41]. The average annual rainfall varies from 600 mm in the east to 2000 mm in the west of the forests (Astara). The main trees of these forests are broadleaf species. A small number of conifers are also found in these forests, for example *Taxus baccata* and *Cupressus* [43]. In northern Iran, the close-to-nature forest management process implemented has led to the development of typical uneven-aged, heterogeneous mixed forest within Hyrcanin forests [44].

In this study, we analyzed eight forest sites in different locations from east to west of the Caspian Sea. The forest sites were selected to cover all Hyrcanian forests from west to east and the plots are well distributed throughout the forest (Figure 1). More details of each forest site given below:

- Nav: The climate of the region is in the humid group based on the humidity coefficient. Based on 10-year statistics (2006–2016), the annual rainfall is 924 mm and the average annual temperature is about 10.2 °C. [45]. The soil type is generally in most parts of forest browns with acidic pH (5.6 to 6.3), loamy to loamy sandy soil texture with good drainage and humus [45].
- Chafroud: The general direction of this forest is north, located at an elevation of 300 to 700 m above sea level. Slope ranges from 20–50%. The type of bed rock is impure limestone, a mixture of igneous rocks with conglomerate and brown to black schist layers, and the soil type is brown forest soil [46].
- Loveh: Loveh forests are located in Golestan province 24 km east of Galikesh city and at an elevation of 700–1900 m above sea level. The average annual temperature is 12.2 °C and the average annual rainfall in the region is 524 mm. The main types of soils are forest brown, Soil pH is between 6.8 and 8.7 [47].
- Ramsar: This forest is located in the 30th watershed, 5th row of Safarood Ramsar in northern Iran, with a range of elevation from 1200 to 1300 m above sea level. The most prevalent soil in the southern part of the study area is alluvial in origin. The average annual temperature is 15.9 °C, the average annual relative humidity is 84.7%, and the average annual precipitation is 1162 mm. [48].
- Sardabroud: The minimum altitude of this forest is 50 m and the maximum is 1400 m. Most precipitation is in the form of rain and most of it is in autumn. The soils of the study area are mainly undeveloped forest brown runes with acidic pH and washed brown with argillic horizon. [49].
- Kheyroud: Gorazbon section with an area of 934.24 hectares, the third section of Kheyroud forest, which is located 7 km east of Nowshahr city. The amount of annual rainfall in Kheyroud region is 1300 mm. The average annual temperature is 15.9 °C [42].
- Haftkhal: This site is located in Parcel 36 Series 4 Section 2 of Neka Forest. The minimum elevation is 900 m and the maximum elevation is 2060 m. Its annual rainfall is about 619 mm and the annual temperature fluctuates between zero and 29 degrees Celsius. The bed rock is limestone and dolomitic limestone, type of brown soil, slightly heavy soil texture. [48].
- Sangdeh: The elevation of this site is about 1280 to 1700 m above sea level and the general aspect of its slope is northeast. Soil type is acidic forest brown. The average annual rainfall is about 845 mm and the average annual temperature is about 11 degrees Celsius. [50].

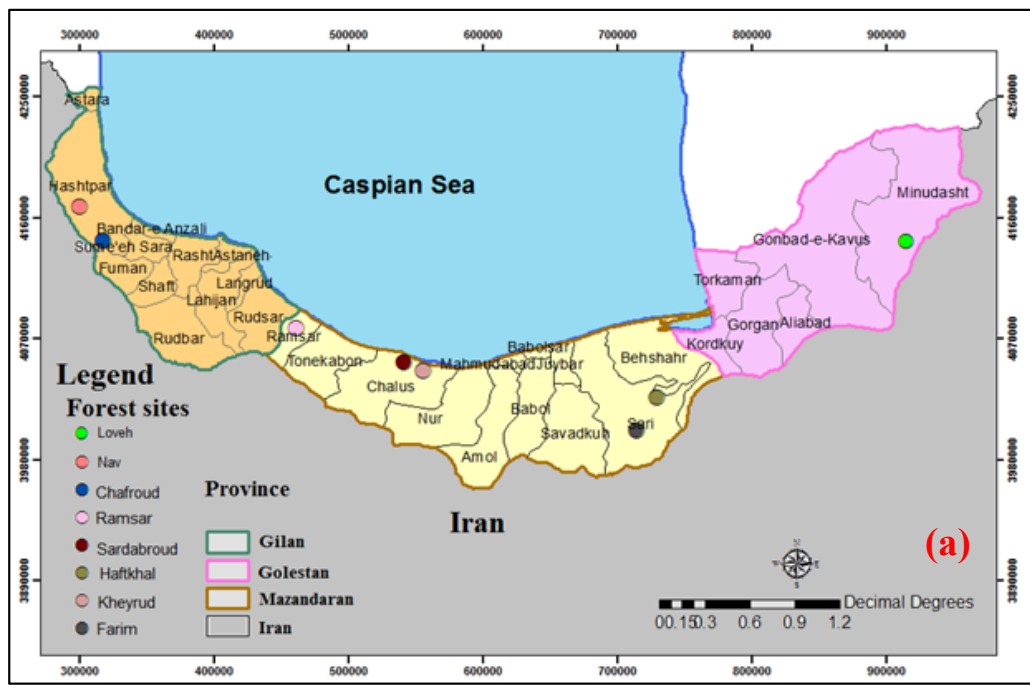

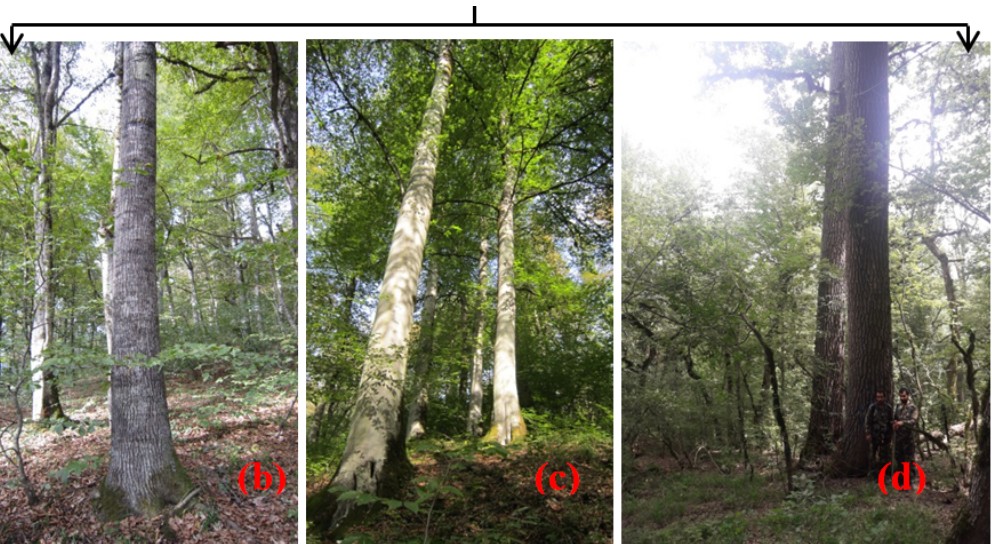

**Figure 1.** Overview of the forest sites in northern Iran and approximate location of them used in this analysis (**a**); *Alnus subcordata*, one of the main tree species on the forest site of the Nav (**b**); *Fagus orientalis*, the main tree species on the forest site of the Kheyroud forest (**c**); *Quercus castaneifolia*, one of the main tree species on the forest site of the Loveh (**d**).

*2.2. Data Collection*

In this study, 15,988 trees were measured in 655 circular permanent sample plots (0.1 ha). In all forest sites, the same protocol was used for data collection, which is as follows:

- In each forest site, a rectangular grid of 200 × 150 m was established in the forest area.
- The DBH of all live trees with a diameter >7.5 cm was measured in the field using a caliper and recorded by species and site data, including slope, aspect, etc. The distance and azimuth between the trees in each sample plot relative to the center of the sample plot were measured and recorded. Average slope, main aspect and average elevation were measured by Suunto clinometer, Vertex or GPS. The geographical coordinates in the center of each plot were determined by GPS [51].

- Due to the mixed and heterogeneous forest stands, and in order to prepare the height curve in each plot, the tree closest to the center of the sample plot and the thickest tree were measured for height and diameter [24].

Table 1 shows descriptive statistics of data used in the tree diversity model. Figure 2 shows the flowchart of main stages of the research.

**Table 1.** Characteristics of all variables on permanent forest monitoring plots in each studied forest.

| Variables | | Forest | | | | | | | |
|---|---|---|---|---|---|---|---|---|---|
| | | Sangdeh | Haftkhal | Kheyroud | Sardabroud | Ramsar | Loveh | Chafroud | Nav |
| Average of DBH (cm) | Mean | 43.15 | 26.56 | 35.85 | 32.17 | 34.74 | 25.83 | 35.51 | 36.89 |
| | Std. Deviation | 13.30 | 8.10 | 16.37 | 6.19 | 12.20 | 5.87 | 12.28 | 9.37 |
| | Minimum | 15.20 | 14.50 | 14.20 | 15.00 | 13.90 | 16.50 | 17.60 | 19.50 |
| | Maximum | 91.50 | 45.30 | 175.00 | 44.60 | 55.70 | 45.30 | 67.80 | 52.90 |
| Sum of BA (m$^2$ Ha$^{-1}$) | Mean | 10.05 | 31.44 | 36.06 | 33.23 | 34.88 | 30.34 | 25.97 | 33.35 |
| | Std. Deviation | 4.19 | 10.36 | 13.97 | 9.26 | 12.66 | 9.25 | 8.44 | 9.58 |
| | Minimum | 0.60 | 1.10 | 4.40 | 16.30 | 5.80 | 16.80 | 7.40 | 19.00 |
| | Maximum | 23.20 | 43.30 | 112.80 | 50.10 | 49.10 | 62.40 | 50.90 | 49.70 |
| Average of BAL (m$^2$ Ha$^{-1}$) | Mean | 5.13 | 27.74 | 32.57 | 24.61 | 29.00 | 23.17 | 21.52 | 27.93 |
| | Std. Deviation | 5.55 | 10.75 | 15.34 | 7.89 | 9.60 | 5.96 | 7.69 | 9.87 |
| | Minimum | 0.10 | 10.80 | 2.20 | 9.20 | 3.80 | 11.70 | 9.00 | 14.40 |
| | Maximum | 78.70 | 57.70 | 112.70 | 40.40 | 43.60 | 43.00 | 42.00 | 52.90 |
| Slope (%) | Mean | 32.30 | 25.85 | 26.04 | 31.32 | 33.33 | 26.55 | 47.62 | 43.21 |
| | Std. Deviation | 13.21 | 14.58 | 15.44 | 21.93 | 18.96 | 11.19 | 17.83 | 13.62 |
| | Minimum | 5.00 | 5.00 | 0.00 | 5.00 | 10.00 | 5.00 | 5.00 | 10.00 |
| | Maximum | 70.00 | 60.00 | 100.00 | 100.00 | 75.00 | 50.00 | 85.00 | 70.00 |
| Elevation (m) | Mean | 1168.30 | 1489 | 1200.30 | 1473 | 161.80 | 1557.00 | 1069.40 | 1653.40 |
| | Std. Deviation | 233.66 | 44 | 81.53 | 41 | 9.84 | 67.05 | 121.31 | 116.35 |
| | Minimum | 815.00 | 1415 | 971.00 | 1407 | 148.00 | 1453.00 | 865.00 | 1469.00 |
| | Maximum | 1650.00 | 1547 | 1342.00 | 1534 | 183.00 | 1686.00 | 1274.00 | 1851.00 |
| Aspect | Mean | 3.58 | 1.65 | 1.72 | 3.07 | 4.20 | 1.41 | 2.48 | 2.64 |
| | Std. Deviation | 2.35 | 0.81 | 1.70 | 2.60 | 2.48 | 1.55 | 2.23 | 0.73 |
| | Minimum | 0.00 | 1.00 | 1.00 | 1.00 | 1.00 | 1.00 | 1.00 | 1.00 |
| | Maximum | 8.00 | 3.00 | 6.00 | 9.00 | 7.00 | 7.00 | 7.00 | 4.00 |
| No of Trees per ha | Mean | 154 | 501 | 421 | 298.75 | 215 | 417.6 | 233.21 | 238 |
| | Std. Deviation | 74.42 | 202.01 | 241 | 76.88 | 76.50 | 160 | 140.12 | 90.75 |
| | Minimum | 30 | 210 | 20 | 190 | 155 | 140 | 40 | 110 |
| | Maximum | 500 | 1080 | 1220 | 410 | 419 | 900 | 630 | 440 |
| Shannon Weiner | Mean | 0.6653 | 0.1476 | 0.6807 | 0.8464 | 0.6819 | 1.1420 | 0.7846 | 0.4392 |
| | Std. Deviation | 0.3675 | 0.1913 | 0.3605 | 0.2815 | 0.2921 | 0.3420 | 0.4660 | 0.3640 |
| | Minimum | 0.0000 | 0.0000 | 0.0000 | 0.4000 | 0.2100 | 0.2900 | 0.0000 | 0.0000 |
| | Maximum | 1.4900 | 0.6000 | 1.5600 | 1.4800 | 1.1700 | 1.6300 | 1.6500 | 1.5200 |

### 2.3. Abiotic Variables

Physiographic factors:

Physiographic factors (elevation, slope and aspect) have a significant effect on the vegetation of an area, and the elevation is a determining factor in temperature and climate regime of an area. At a specified elevation, the slope and direction of the slope affect the light in an area and influence the micro-environment [52].

### 2.4. Biotic Variables

- Average DBH in plot:

Diameter at breast height (DBH), is an important factor and the foundation for modeling forest growth and yield [53]. This variable is relatively easy to measure so associated data is abundant [54].

- Basal area (BA) (m$^2$ ha$^{-1}$)

Basal area is area defined as the sum of the cross section areas of trees measured at DBH and expressed on a per hectare basis. Basal area per unit area is an important

indicator in forest management and ecology because it is closely related to the volume of forest stands. [55].

- Basal area in largest trees (BAL) (m² ha⁻¹**)**

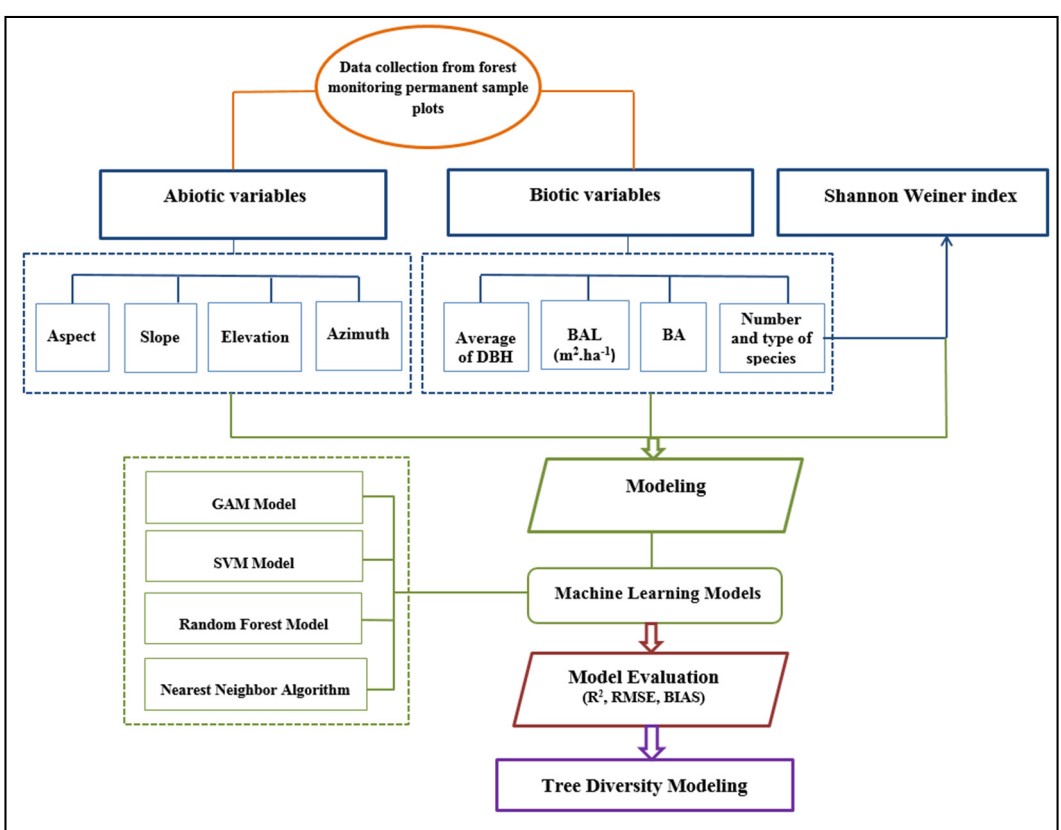

**Figure 2.** Flowchart of main stages of the research.

BAL is important because it is related to the light available to trees. The amount of light available to small trees decreases with increasing BAL. This index is also important in determining the competition in plots, as it is a sign of the dominance of trees in a stand. [56]. BAL was calculated as follows (Equation (1)):

$$BAL = \frac{\pi}{4} \times \sum_{j=1}^{n} \left( TF_j \times DBH_j^2 \right) \tag{1}$$

where $DBH_j > DBH_i$ (i.e., all trees larger than subject tree *i*), *DBH* is measured in cm and $TF_j$ is a tree factor, i.e., the number of trees represented by *j*th tree in a hectare derived from area) [57].

- Number of trees per hectare

Number of tree an important indicator in forest measurement and shows the relative number of trees in an area or a stand. This index is an indicator of the structure of a forest stand and is very important in this regard [58]. Number and type of species is measured as follows (Equation (2)):

$$N = \frac{\overline{n}}{a} \tag{2}$$

where, *N* is number of tree per hectare, $\overline{n}$ is average number of trees in plots and *a* is the area of plots per hectare.

- Tree Species

In this study, tree species are used as bio-variables in the models. The total species are divided into 6 groups in the area, which include the following: *Fagus orientalis* Lipsky, *Carpinus betulus* L., *Quercus castaeifolia*, *Alnus subcordata*, *Acer velutinum* Boiss and others species.

### 2.5. Shannon Weiner Index

Indices of species diversity are in fact a combination of species richness and uniformity. Shannon Weiner index adds the two values of species richness and uniformity in one quantity [24]. In this study, Shannon Wiener indices were used to investigate species diversity. For estimating this index, 15,988 trees were measured in 655 plots and the species was identified. This index is obtained as follows (Equation (3)):

$$H = -\Sigma pi \ln(pi). \tag{3}$$

where H is Shannon Wiener index and pi is the relative abundance of ith type.

### 2.6. Statistical Analysis

In order to compare the eight study areas, in terms of biodiversity indices, after achieving the conditions of normality and homogeneity of variance, one-way ANOVA and Tukey and Duncan tests were used for general comparison between groups and mean comparison, respectively. All of the above calculations were performed using SPSS 22 software, the results of which are presented in Tables 2 and 3.

**Table 2.** Results of analysis of variance between eight forest sites studied.

|  | Sum of Squares | Df | Mean Square | F | Sig. |
|---|---|---|---|---|---|
| Between Groups | 14.663 | 7 | 2.095 | 16.117 | <0.001 |
| Within Groups | 85.259 | 648 | 0.130 |  |  |
| Total | 99.922 | 655 |  |  |  |

**Table 3.** Tukey and Duncan test results were computed among eight forest sites.

|  | Group | N | Group | | | |
|---|---|---|---|---|---|---|
|  |  |  | Nave | Chafroud | Loveh | Ramsar |
| Tukey HSD | Haftkhal | 20 | 0.147620 |  |  |  |
|  | Nave | 28 |  | 0.439150 |  |  |
|  | Sangdeh | 313 |  | 0.665299 | 0.665299 |  |
|  | Kheyroud | 202 |  | 0.680663 | 0.680663 |  |
|  | Ramsar | 15 |  | 0.681927 | 0.681927 |  |
|  | Chafroud | 29 |  |  | 0.784590 |  |
|  | Sardabroud | 28 |  |  | 0.846354 |  |
|  | Loveh | 29 |  |  |  | $1.142786 \times 10$ |
|  | Sig. |  | 1.000 | 0.152 | 0.517 | 1.000 |
| Duncan | Haftkhal | 20 | 0.147620 |  |  |  |
|  | Nave | 28 |  | 0.439150 |  |  |
|  | Sangdeh | 313 |  |  | 0.665299 |  |
|  | Kheyroud | 202 |  |  | 0.680663 |  |
|  | Ramsar | 15 |  |  | 0.681927 |  |
|  | Chafroud | 29 |  |  | 0.784590 |  |
|  | Sardabroud | 28 |  |  | 0.846354 |  |
|  | Loveh | 29 |  |  |  | $1.142786 \times 10$ |
|  | Sig. |  | 1.000 | 1.000 | 0.083 | 1.000 |

*2.7. Modeling Approaches*

2.7.1. GAMs Models

The generalized additive model is a nonparametric generalized extension of the linear model. In generalized additive models, unlike generalized linear models, the data determine the shape of the response curve. GAMs provides useful information to analyze ecological data and determine nonlinear relationship between different variables. The important thing about these models is that they are data-driven instead of axial [59].

2.7.2. SVM Models

Support vector machine is a nonparametric method and binary classifier [60]. In this study, four kernel types were examined, namely linear, sigmoidal, polynomial and RBF-based type [61]. The kernel parameters include gamma ($\gamma$), epsilon ($\varepsilon$), and capacity (c). To evaluate the model fits, gamma values were calculated as the reciprocal of a number of independent variables [62]. The specified grid search included a range of capacity from 1 to 50, which is equal to the range of input variables [63]; epsilon rates varied from 0.1 to 0.5. Internally, the SVM calculates the model to optimize classification. The strictness of this optimization results is controlled by the capacity and epsilon parameters.

2.7.3. Random Forest Models

The RF model is a supervised machine-learning algorithm based on decision trees that involves a multitude of classification and regression trees [64]. The most important feature of stochastic forests is their high performance in measuring the importance of variables to determine what role each variable plays in predicting the response.

The random forest algorithm is based on a set of decision trees and is currently one of the best learning algorithms [25]. The random forest model is based on averaging the results of all decision trees and performs high-accuracy classification for many datasets. We used 655 initial trees for training and testing and created twelve nodes to produce a graph that shows the average squared error rate against each tree based on our input data. One of the most important parameters that must be set is the k-predictor for each node. Calculation of the square root of the total number of independent variables is the simplest way to determine k.

2.7.4. KNN Models

K-nearest-neighbor (KNN) regression, one of the most popular data-mining algorithms, has been used to solve generalized linear modeling problems [65], particularly for classification problems [66].

This method is a suitable tool for estimating forest information. To run the algorithm, three parameters are important: number of neighbors, distance, and weighting (including weighting of the nearest neighbors). One of the parameters that must be determined in this method is the type of distance. The four-distance criteria Euclidean, Euclidean Square, Manhattan and Chebychev were weighted and according to the weighting of the variables, the data were used as standardized [67].

*2.8. Model Fitting and Evaluation*

For each bootstrap analysis, the ground observation data were divided randomly into a training set (70% of the data) and a validation set (30% of the data). We used several statistical measures to quantify model performance [49,50], including the $R^2$, (Equation (4)), RMSE (Equation (5)), and mean residual deviation (bias) (Equation (6)).

$$R^2 = \frac{\text{Regression SS}}{\text{Total SS}} = \frac{\sum (yi - \hat{yi})^2}{\sum (yi - \overline{y})^2} \tag{4}$$

$$\text{RMSE} = \sqrt{\frac{1}{n}\sum_{i=1}^{n} yi - \overline{y}}^{\,2} \tag{5}$$

$$\text{Bias} = \frac{1}{n} \sum\nolimits_{(i=1)}^{n} y_i - \bar{y} \tag{6}$$

where $y_i$ is estimated tree diversity, $\bar{y}$. is the mean of observed tree diversity, and $\hat{y}i$ is the mean of the estimated tree diversity. It should be noted that the two data sets should be similar in terms of their standard deviation and average deviations, since they were randomly selected from the original population of data. Before feeding data to R and STATISTICA software, data sets were normalized within the range of $-0.9$ and $0.9$ to increase the effectiveness of the training process and the overall quality of modeling results [25].

## 3. Results

In this study, the Shannon Wiener diversity index was calculated for the eight forest sites. Tables 2 and 3 show the results of the analysis of variance between the eight forest sites, and, as can be seen, there are significant differences between the regions in terms of their Shannon diversity index This index ranged from 1.142 in Loveh forest in Golestan province to 0.147 in the Haftkhal forest in the east of Mazandaran province. Also, the analysis of variance test showed a significant difference between the eight regions in terms of Shannon diversity index, according to the results of Tukey and Duncan tests, site 3 (Loveh forest in Golestan province) is the most different from the other forest sites.

In Figure 3, the highest Shannon diversity index is in Loveh forest site (1.142) and the lowest is related to Haftkhal forest site (0.147).

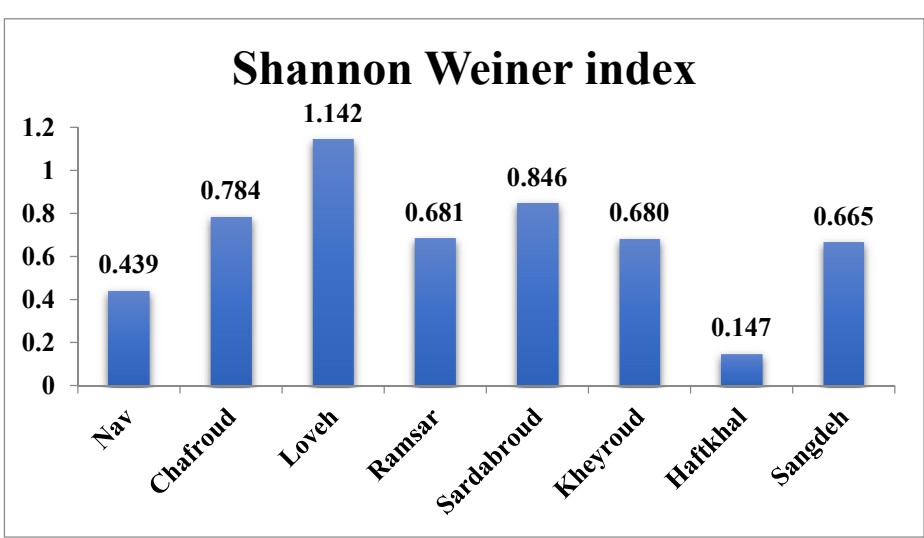

**Figure 3.** Average Shannon Biodiversity Index calculated for each forest site in the permanent sample plots (0.1 ha).

### 3.1. Generalized Additive Model (GAM)

The following equation presents the relationship of the Shannon diversity index with BA and elevation:

$$H = 0.63 + 0.007\,(BA) - 0.0008\,(Elevation) \tag{7}$$

where $H$ is the Shannon diversity index, $BA$ is the basal area (m$^2$ ha$^{-1}$) and *Elevation* is height above sea level (m). The rest of the biometric and physiographic parameters were not significant in the equation. BA (t = 2.51, *p* = 0.02) and elevation (t = $-0.05$, *p* = 0.00) were statistically significant, resulting in an $R^2$ of 0.17, a RMSE of 0.47 and mean residual deviation (Bias) of 0.87. The relative RMSE was 70.14%. The residuals showed a positive bias in plots where Shannon index exceeded 0.8 (Figure 4).

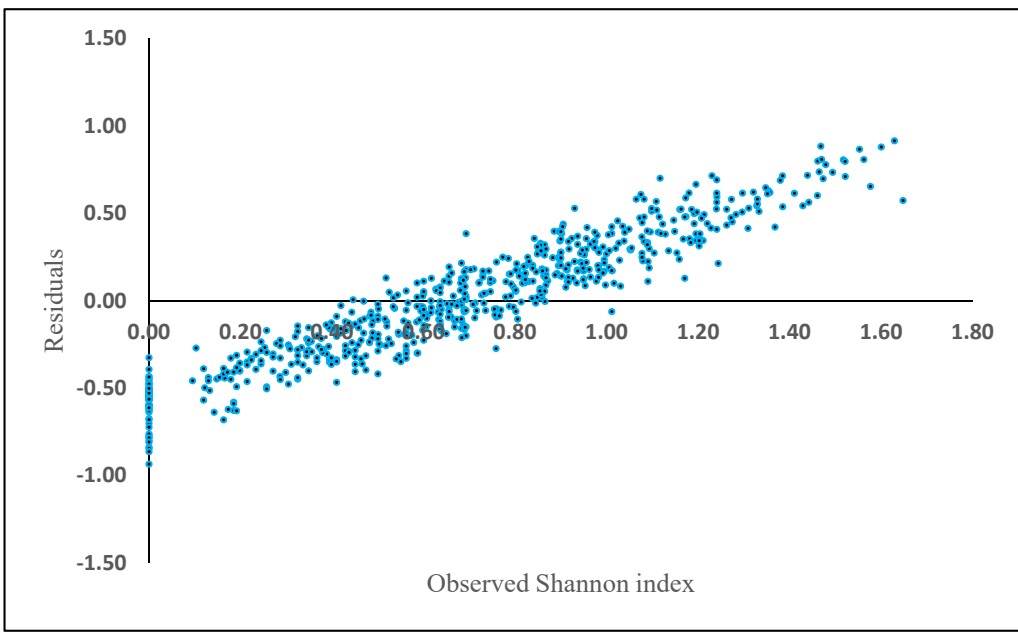

**Figure 4.** Model residuals of Shannon index plotted against observed Shannon index based on the generalized additive model (GAM).

### 3.2. Support Vector Machine

Results showed the RBF Gamma kernel had the smallest RMSE and bias in the training and evaluation models (Table 4).

**Table 4.** Evaluation of SVMs based on different kernel types.

| Kernel Type | Linear | | Polynomial | | RBF Gamma | | Sigmoid | |
|---|---|---|---|---|---|---|---|---|
| | **Train** | **Evaluation** | **Train** | **Evaluation** | **Train** | **Evaluation** | **Train** | **Evaluation** |
| $R^2$ | 0.30 | 0.32 | 0.34 | 0.34 | 0.40 | 0.41 | 0.28 | 0.1531 |
| RMSE | 0.41 | 0.39 | 0.39 | 0.37 | 0.33 | 0.32 | 0.47 | 0.41 |
| BIAS | 0.12 | 0.12 | 0.10 | 0.10 | 0.03 | 0.02 | 0.19 | 0.16 |
| %RMSE | 61.19 | 58.20 | 58.20 | 55.22 | 49.25 | 47.76 | 70.14 | 61.19 |
| %BIAS | 17.91 | 17.91 | 14.92 | 14.92 | 4.47 | 2.98 | 28.35 | 23.88 |
| Gamma | - | | 0.08 | | 0.08 | | 0.08 | |
| Epsilon | 0.1 | | 0.1 | | 0.1 | | 0.1 | |
| Capacity | 10 | | 10 | | 10 | | 10 | |

Elevation (230.14), Aspect (123.55), Slope (120.99), basal area (82.06), BAL (66.75), diameter (58.90), Fagus (55.08), Carpinus (50.01), others species (32), Alnus (19), Quercus (9) and Acer (3) are the total weight of each variable separately obtained in the model. The residuals showed a positive bias in plots where Shannon index exceeded 0.75 (Figure 5).

### 3.3. Random Forest

The Shannon index using the random forest method had an "$R^2$" of 0.59, with a RMSE of 0.28, a BIAS of 0.01, respectively. The residuals showed a positive bias in plots where Shannon index exceeded 0.75 (Figure 6), similar to the previous analyses.

Decision trees are useful tools for classifying and predicting data. The final tree is made up of many nodes connected by branches. The nodes at the end of the branches are called leaf nodes and play a special role in forecasting. In Figure 7, each node contains information about the number of samples (N) in which the node and the distribution of the values of the variables are dependent. The information in the root node is all the information in the training data. The final decision tree of the random forest classification method has

12 non-terminal children and 13 terminal nodes. Elevation is the most important variable in this method and the other variables are as follows: elevation (importance = 1.0), Basal area (0.73), BAL (0.64), diameter (0.63), slope (0.48), aspect (0.46), Fagus (0.35), Carpinus (0.25), others species (0.19), Alnus (0.07), Quercus (0.05) and acer (0.02). Also, based on the results, modeling with seven variables in each node (k = 8) resulted in a minimum of RMSE and was selected as optimal K for estimating the net annual volume increment.

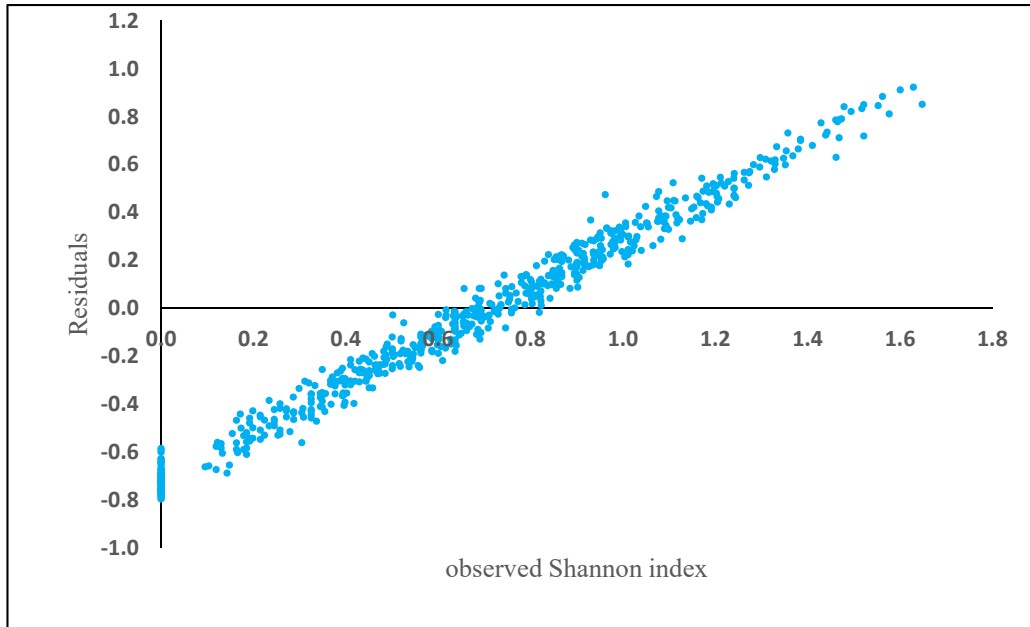

**Figure 5.** Model residuals of Shannon index plotted against observed Shannon index based on the best SVM algorithm.

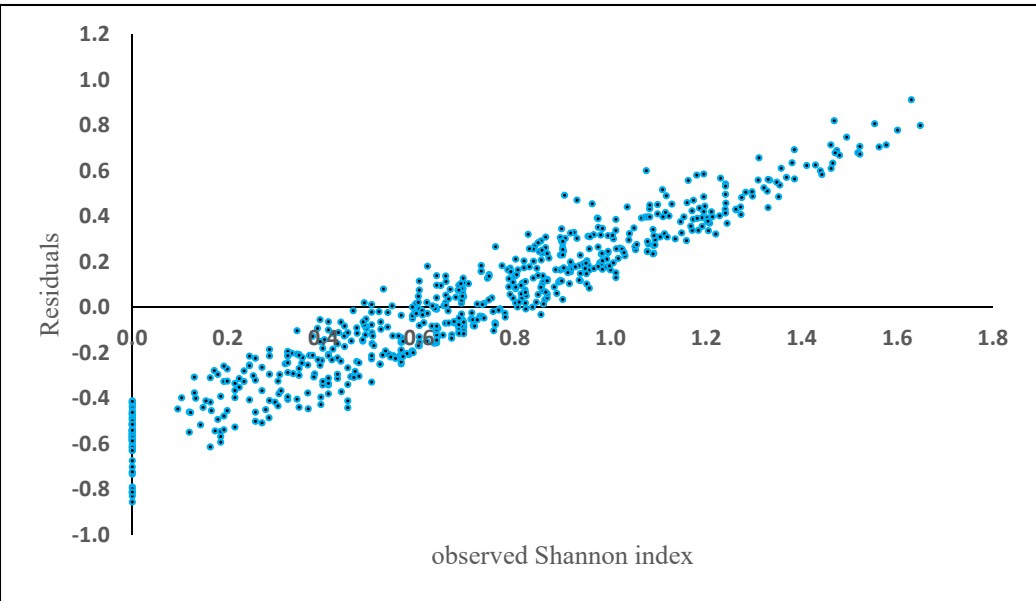

**Figure 6.** Model residuals of Shannon index plotted against observed Shannon index based on the Random Forest results.

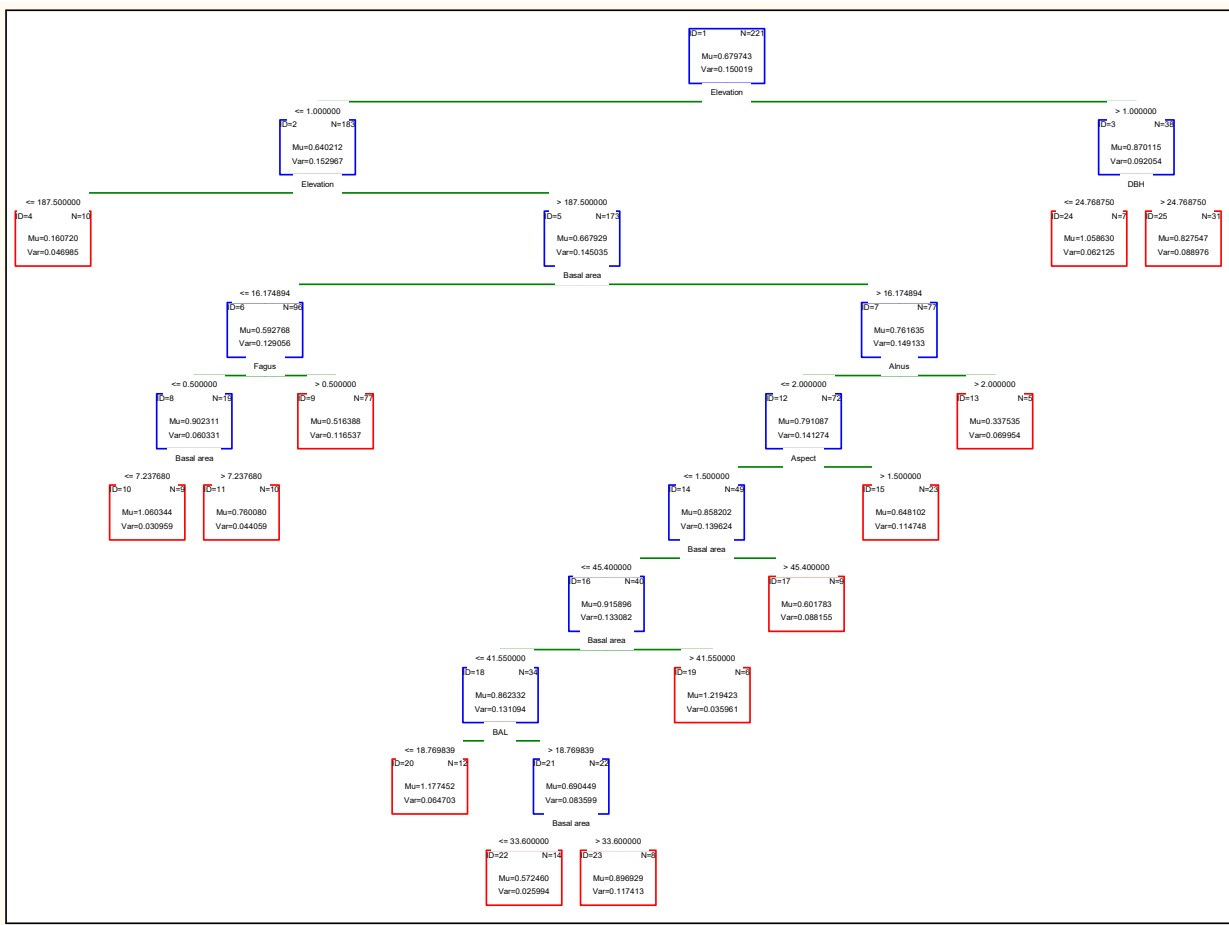

**Figure 7.** The decision tree derived from Random Forest.

Figure 8 shows results of modeling with RF model. As shown in Figure 8, the RF model (which was recognized as the best ML model), were indicated the most influential factors defining variation of tree diversity, respectively. Elevation, BA and BAL were the most biotic an abiotic factors.

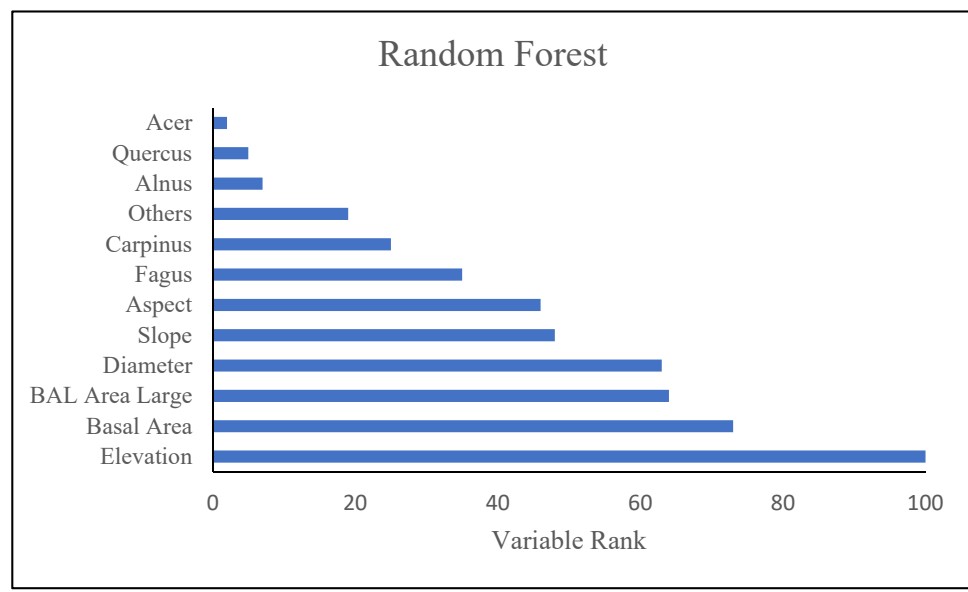

**Figure 8.** Relative importance of predictor variables in the Random Forest (the best model of machine learning).

### 3.4. Nearest Neighbor Algorithm

The nearest neighbor algorithm, weighted with a range of 1 to 20 and each of four distance metrics, showed that all methods have almost the same results and among them the Manhattan distance metric with K = 11 neighbors resulted in the smallest RMSE and bias for both training and evaluation datasets (Table 5). Even with the best model, the relationship between estimated and observed Shannon index was modest, resulting in an $R^2$ of 0.30.

**Table 5.** Results and Evaluation Related to Analysis of K-Nearest Neighbor.

| KNN | Euclidean | | Euclidean Squared | | Manhattan | | Chebychev | |
|---|---|---|---|---|---|---|---|---|
| | **Train** | **Evaluation** | **Train** | **Evaluation** | **Train** | **Evaluation** | **Train** | **Evaluation** |
| K range | 1–20 | 1–20 | 1–20 | 1–20 | 1–20 | 1–20 | 1–20 | 1–20 |
| $R^2$ | 0.28 | 0.28 | 0.28 | 0.28 | 0.29 | 0.30 | 0.29 | 0.29 |
| RMSE | 0.48 | 0.48 | 0.47 | 0.47 | 0.44 | 0.42 | 0.49 | 0.48 |
| BIAS | 0.08 | 0.08 | 0.08 | 0.08 | 0.07 | 0.06 | 0.07 | 0.07 |
| %RMSE | 71.64 | 71.64 | 70.14 | 70.14 | 65.67 | 62.68 | 73.13 | 71.64 |
| %BIAS | 11.94 | 11.94 | 11.94 | 11.94 | 10.44 | 8.95 | 10.44 | 10.44 |
| Optimal K | 10 | 10 | 10 | 10 | 11 | 11 | 10 | 10 |

In contrast to the previous methods, in this method, the variability of bias is high and the range of negative residuals is greater than for the previous methods, which were generally less than −1.0 (Figure 9).

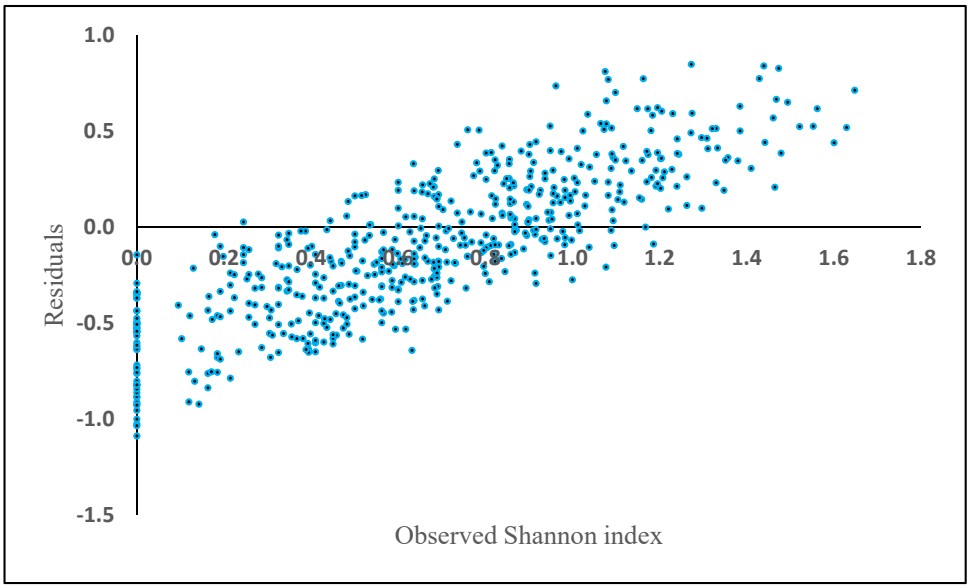

**Figure 9.** Model residuals of Shannon index plotted against observed Shannon index based on the nearest neighbor algorithm.

A comparison of the coefficients of determination, RMSEs, and relative RMSEs of the different estimation methods showed that the random forest method resulted in the best model (Table 6). The models derived from the support vector machine and nearest neighbor algorithm methods resulted in models with lesser fits. The Generalized additive model (GAM) method resulted in the poorest model fit based on $R^2$ and the greatest RMSE value of all methods that were evaluated.

**Table 6.** Comparison of fit statistics of the methods tested in this study.

| Method | $R^2$ | RMSE | %RMSE |
|---|---|---|---|
| Random Forest (RF) | 0.59 | 0.28 | 41.79 |
| Support Vector Machine (SVM) | 0.41 | 0.32 | 47.76 |
| Nearest Neighbor Algorithm (KNN) | 0.30 | 0.42 | 62.68 |
| Generalized additive model (GAM) | 0.17 | 0.47 | 70.14 |

## 4. Discussion

### 4.1. Trend of Tree Diversity in the Hyrcanian Forest

The diversity of tree species is the basis of biodiversity of the whole forest, because trees provide resources and habitats for other forest species. Species diversity in the forest changes under the influence of various factors [42]. Biodiversity ensures flexibility and adaptability of forest ecosystems, which protects the environment and leads to sustainable forest management [68]. Currently, considering biodiversity in forest management, along with other accepted economic and environmental criteria, it is believed that in order to achieve the goals of sustainable forest management, forestry activities must be in line with environmental issues, especially plant biodiversity [69]. Also, the use of biodiversity indicators to evaluate different functions that have been developed for ecological indicators allows study of environmental characteristics, forest management [70,71], and conservation [72] as applied to ecosystems. Because plants are the result of the environmental characteristics of each region, they are a full-fledged mirror of the habitat characteristics of that region [73]. Therefore, the study of plant composition and plant biodiversity can be used as an appropriate guide in ecological evaluations and study of biodiversity in each region. According to the results of the present study, biodiversity from west (Gilan Province) to east (Golestan Province) often has an increasing trend in the Hyrcanian forest. Temperature increases from west to east as does biodiversity [23], But the annual precipitation decreases from 1345 mm at the Asalem forest in Gilan Province at the westernmost point to 524 mm at the Loveh forest in Golestan Province in the easternmost point. In addition, there are other factors that increase or decrease biodiversity in a forest site, the most important of which is elevation, Species diversity decreases with increasing elevation. But there are exceptions to this trend seen in this study; Loveh forest, despite being located in the easternmost point of the study, had the highest Shannon index (1.142) due to the predominance of oak species in this forest. Because it is a shade-intolerant species (unlike beech, which is shade-tolerant), it has more species diversity than the forests in which beech is predominant.

Shannon index in the Nav forest was 0.439, which is the lowest value after Haftkhal forest. Despite this site being in the west of Gilan province, and the expectation of a high index of diversity, and in line with the general trend of biodiversity from west to east in the Hyrcanian forest, this decrease can be considered to result from intense exploitation of these forests.

Shannon index of 0.784 was relatively high on the Chafroud site, located in the west of the Hyrcanian forests and in the middle highlands.

In Mazandaran Province, from west to east, respectively, the index for Ramsar, Sardabroud, Kheyroud and Haftkhal forest sites was 0.846, 0.681, 0.680 and 0.147, respectively. The trend of diversity from west to east was consistent, with the exception of the Sardabroud site, which, being at the mid elevation level, showed high biodiversity index. On the other hand, HaftKhal forest site in Neka city had the lowest Shannon index, which apparently is related to both natural causes and management factors.

Extreme degradation and exploitation in the Neka region has led to high dominance for species such as beech, resulting in high dominance index and very low uniformity. In most of the sample plots, one or two dominant species with high frequency (beech species) were present and the presence of other species was low or not observed at all, which caused a decrease in uniformity and increased dominance in this area. In addition, intense management and exploitation play an important role in relation to the low diversity index in the

Neka forest area, and another important factor in this regard is environmental influences, including elevation. This site is located at a relatively high elevation as compared to the other two areas. This, as well as the dominance of the northern direction in this region, can be another factor for the dominance of beech species and overcoming competition with other species.

*4.2. Machine Learning Approach to Modeling Diversity*

In this research, four ML models were implemented including generalized additive model (GAMs), support vector machine (SVM), random forest (RF) and the K-nearest neighbor algorithm (KNN). Among ML models, The RF model with $R^2$ 0.59 and RMSE 0.28 was the best model followed by SVM, the nearest neighbor algorithm, and GAM models with $R^2$ 0.41, 0.30 and 0.17, respectively. Thus, as can be seen, the RF model was the best model among the ML models, which was similar to the results of many studies in this field [25,74]. The reason for the superiority of the RF model over other models can be that this model is more powerful than other machine learning methods in reducing data and that it is also less sensitive to parameter adjustment and is able to evaluate the importance of variables. In addition, it generates an internal unbiased estimate of the generalization error as the model building progresses [35,36]. In a study similar to ours support vector regression (SVR), modified regression trees (RT) and random forest (RF) were used in determining forest stand height using plot-based observations and airborne LiDAR data [75]. Similar to the present study, which considers the efficiency of SVM and RF models in modeling forest biodiversity, they concluded that there was no statistically significant difference in plot height estimation between these models and all of them are acceptable. Also, in a study in Western Himalaya [76] that used machine learning methods including classification regression tree (CART), random forest (RF), and support vector machine (SVM) algorithms showed results similar to the present study; the authors concluded that RF model has a higher accuracy in forest fire burn area. A study in Tasmania, Australia [77] used support vector regression (SVR), artificial neural networks (ANN), random forest (RF), and gradient boosted regression trees (GBRT) for mapping forest cover and exploring influential factors, and their findings were in line with the results of our study. In terms of projection accuracy, and required less computational costs, RF far outperformed the other three models [77].

*4.3. Effect of Biotic and Abiotic Factors on Biodiversity Index*

The distinguishing feature of this study is the simultaneous consideration of the most important biotic and abiotic factors on biodiversity, which in comparison with related studies is one a few of studies that does so. Physiography affects biodiversity of the forest. According to the results obtained from the models, especially the random forest model, the effect of elevation was one of the most important factors affecting the biodiversity of forests in northern Iran. Changes in elevation, microclimatic, ecological and environmental conditions of the forest habitat, and the structural condition of the area changes in proportion to the local conditions [42]. With the change of the conditions governing the habitat, tree diversity changes and increases in the favorable ecological and structural conditions of the land, and their amount decreases in unfavorable conditions. In the Haft Khal region, due to being located at a higher altitude, there is less diversity in the region. In addition, studies in USA (including all states of the USA east of North Dakota, South Dakota, Nebraska, Kansas, Oklahoma, and Texas) [78] and in 11 remnant grasslands within the Aspen Parkland Ecoregion of central Alberta, in western Canada [79], have stated that elevation is one of the most important factors affecting biodiversity of the region, which is in line with the results of this study. Then biometric factors such as Basal area and Basal area of the thickest trees, physiographic factors such as slope and aspect and finally the type of species were mentioned as factors affecting biodiversity in the region. Species such as *Fagus orientalis* and *Carpinus betulus* are important in the region, while other species also had a positive effect in the modeling process [80]. Similarly, patterns

of diversity across different climate zones [23] in four climate regions in China, including tropical (three sites in Yunnan Province), subtropical (two sites in Hubei Province), warm temperate (one site Gansu Province) and temperate (one site Xinjiang Uygur Autonomous area) showed that both biotic and abiotic factors change the diversity of the forest community between different climatic zones. Other studies have considered both biotic and abiotic factors Assessing the relative importance of mean air temperature, nitrogen availability and direct plant interactions in determining the millennial-scale population dynamics for four temperate tree taxa in the Scottish Highland concluded that all of factors are important [20,21]. Also, as the sensitivity analysis in Figure 8 shows, elevation was the most important factor affecting biodiversity in the selected model (RF). the next factors were BA and BAL, respectively. Tree species was also important factor for the random forest model. Physiographic factors have a significant impact on the biodiversity index in forests, especially the elevation factor. Similar studies have shown that physiographic factors play an important role as indicators of richness and diversity of species [24] determined affecting biotic and abiotic factors. In their study wind, topographic wetness index (TWI) and elevation were most important affecting factors in tree species richness variations. As seen in the results of this study, elevation was the main influential factor in RF models. This variable is a significant predictor affecting species diversity, as observed in previous studies [81,82]. Similarly, aspect, slope and altitudinal variation in Ethiopian landscapes have influenced the existence of varied vegetation types and diversity [83]. By contrast we used modeling to examine the relationship between environmental factors and the biodiversity index while they sampled quadrats and recorded data on species identity, abundance, elevation, slope and aspect. Also, they used different diversity indices and ordination techniques to analyze the data. Furthermore, elevation was one of the most important factors influencing community distribution and species diversity in the Balhus Mountains Reserve of Beijing, China [6], this study examined the functional diversity in the elevation gradient, while we used modeling using ML.

The results of various studies show that the middle elevations generally have the highest index of richness and species diversity, which in the present study is the reason for the high level of these indicators in Sardabroud forest site.

In Haftkhal in Neka forest site, elevation is relatively higher than other areas; this, as well as the dominance of the northern direction in this region, can be another factor for the dominance of the beech species and its dominance in competition with other species. In fact, less uniformity at high elevations is due to the abundance and dominance of beech species.

In general, the results of similar studies show that lower temperatures and slower melting of ice in these areas, especially at high elevations, causes less variety in diversity. These slopes are more humidity and colder causing the dominance of beech species and as a result reduce the uniformity index. Although, it is worth mentioning that in different regions, due to their climatic and geological characteristics and geographical location, different results are obtained about aspect, but usually diversity is greater in aspects with higher humidity and temperature [84].

Another important factor that affects the diversity index, along with natural factors, is management and conservation practices in the forests. In fact, the huge difference in biodiversity index between forest sites with similar elevation and environmental conditions, despite the relative similarity of climatic, physiographic and biological conditions, can be attributed to the management style in these forests.

Therefore, in general, the Shannon diversity index in the managed and protected forest area is significantly larger compared to other areas, which indicates that tree felling and human pressure in the area has resulted in more heterogeneity in the number and diversity of reproduction of different species.

It is necessary to mention that it seems that other environmental factors such as humidity, precipitation, temperature and soil in the studied forest sites, one would expect that the modeling coefficient will increase. Thus, if a model is developed that includes all the other abiotic factors (mentioned), the $R^2$ should be much higher. In our research, results

show the capability of some machine learning techniques to produce accurate estimates of biodiversity index in forest sites and to identify important variables (e.g., elevation, BAL). Although it cannot be said that RF techniques may always be better than other machine learning methods, our results showed a higher coefficient of determination and lower RMSE than other ML methods evaluated. The same results were achieved using ML and geo statistical methods [34] to predict aboveground biomass in Chinese forests, which concluded that the random forest created a reliable and accurate method for AGB mapping in subtropical forest regions with complex topography.

## 5. Conclusions

The main goal of natural resource management is to preserve biodiversity in natural ecosystems, assuming that habitats with more biodiversity have more ecological stability and fertility than areas with less biodiversity, and more stable ecosystems will be more dynamic. In this study, by combining biotic and abiotic factors in ML models, we analyzed their relationship to biodiversity indices across eight forest sites in the Hyrcanian forest in northern Iran. Four machine learning algorithms including GAM, SVM, RF, and nearest neighbor algorithm were used to model tree diversity. The results showed that machine learning methods, especially the random forest and support vector machine, were more accurate than other methods. Based on results of the RF model, elevation, BA, and BAL, were indicated as the most influential factors defining variation of tree diversity in the Hyrcanian forests. Also, in this study, we simultaneously examined the important biotic and abiotic factors in relation to the biodiversity index, which distinguishes this study from similar studies.

Machine learning techniques can often superior to traditional methods when assumptions model for applying parametric procedures are not validated Also, flexibility, accuracy, and the ability to model complex and nonlinear relationships are features of ML methods.

**Author Contributions:** M.B. conceived and designed the experiments; M.B., M.H. and S.H. performed the experiments and analyzed the data; M.B. and S.K.H. contributed reagents/materials/analysis tools; M.B., H.B. and M.N. wrote the paper. All authors have read and agreed to the published version of the manuscript.

**Funding:** This research received no external funding.

**Data Availability Statement:** The data underlying this article will be shared on reasonable request to the corresponding author.

**Acknowledgments:** We acknowledge the anonymous editor and reviewers who provided many helpful comments and suggestions for improving this manuscript.

**Conflicts of Interest:** None of the authors have any actual or potential conflict of interest that could inappropriately influence, or be perceived to influence, this work.

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
