# Peer review of "Assessing Biotic and Abiotic Effects on Biodiversity Index Using Machine Learning"

_forests, doi:10.3390/f12040461_

Round 1
Reviewer 1 Report
Dear authors,
I carefully read the manuscript titled "Assessing biotic and abiotic effects on tree species richness in eight uneven egad and mixed forests using machine learning approach".
In my opinion, the paper is focused on a very interesting topic and is well set, with data measured on 655 plots and nearly 16.000 trees. It is not common to find a dataset with such a large amount of data! The introduction is exhaustive and in general, the work is well organized. However, I have some remarks about the biotic variables used in the models and about the discussion.
In my opinion, biotic variables should include also the main three species of the monitoring plots. You too, in the discussion, state that the predominance of oak species probably affects the value of Shannon Index because oak is a shade-intolerant species and has more species diversity than the forests in which beech is predominant. So why not insert main tree species into the models? They could show that it one of the most important factors affecting tree diversity.
As for the discussion, I think that, unlike the rest of the paper, it is not well written and not well-argued. For example line 408 is materials and methods, lines 409 to 413 are results. In lines 414 - 422 you comment that rainfall and biodiversity decrease from west to east but in the following paragraph (lines 423 - 426) you state the contrary: the highest Shannon index was in the easternmost point of the study.
Together with the criticism I referred above to the models, I think that the low quality of this section affects the suitability of your paper. I see also that very few references are reported in this section. I suggest you re-writing this section discussing your results in light of literature researches.
For the above reasons, in my opinion, this paper, in the present form, is not acceptable for publication in Forests, and it should be subjected to major revision.
Just a minor observation: there is a mistake in the title: “…uneven aged and mixed forests…” not “…uneven egad and mixed forests…”
Author Response
Dear Dr Editor in chief and Dear Reviewers
Thank you for considering a revised version of our previous paper " Assessing biotic and abiotic effects on tree species richness in eight uneven aged and mixed forests using a machine learning approach” (Manuscript ID: forests-1159464) that incorporates the many useful comments and suggestions of the reviewers. Detailed corrections are listed below point by point. Based on the reviewer comments, we have substantially modified the paper.
Yours sincerely,
comment: In my opinion, biotic variables should include also the main three species of the monitoring plots. You too, in the discussion, state that the predominance of oak species probably affects the value of Shannon Index because oak is a shade-intolerant species and has more species diversity than the forests in which beech is predominant. So why not insert main tree species into the models? They could show that it one of the most important factors affecting tree diversity.
Response:
We revised the manuscript as suggested The type of species was added as a variable.
Comment: As for the discussion, I think that, unlike the rest of the paper, it is not well written and not well-argued. For example line 408 is materials and methods, lines 409 to 413 are results. In lines 414 - 422 you comment that rainfall and biodiversity decrease from west to east but in the following paragraph (lines 423 - 426) you state the contrary: the highest Shannon index was in the easternmost point of the study.
Response: We re-write discussion section as suggested and revised these paragraphs.
4.1. Trend of tree diversity in the Hyrcanian forest
The Hyrcanian forest provides a wide range of ecosystem services of great value, including landscape, climate and water regulation, supporting atmospheric quality, timber and non-timber forest products, opportunities for tourism, recreation, health and wellbeing and spiritual experiences. The diversity of tree species is the basis of the biodiversity of the whole forest, because trees provide resources and habitats for other forest species. Species diversity in the forest changes under the influence of various factors [42]. Biodiversity ensures flexibility and adaptability of forest ecosystems, which protects the environment and leads to sustainable forest management [68]. Currently, considering biodiversity in forest management, along with other accepted economic and environmental criteria in the world, it is believed that in order to achieve the goals of sustainable forest management, forestry activities must be in line with environmental issues, especially plant biodiversity [69]. Also, the use of biodiversity indicators to evaluate different functions that have been developed for ecological indicators allows study of environmental characteristics, forest management [70; 71), and conservation [72] as applied to ecosystems. Because plants are the result of the environmental characteristics of each region, they are a full-fledged mirror of the habitat characteristics of that region [73]. Therefore, the study of plant composition and plant biodiversity can be used as an appropriate guide in ecological evaluations and study of biodiversity in each region. According to the results of the present study, biodiversity from west (Gilan province) to east (Golestan province) often has an increasing trend in the Hyrcanian forest, this is an important factor in creating this trend. In addition, temperature increases from west to east as does biodiversity [23]. But the annual precipitation decreases from 1345 mm at the Asalem forest in Gilan province at the westernmost point to 524 mm at the Loveh forest in Golestan province in the easternmost point. In addition, there are other factors that increase or decrease biodiversity in a forest site, the most important of which is elevation, Species diversity decreases with increasing elevation. But there are exceptions to this trend seen in this study; Loveh forest, despite being located in the easternmost point of the study, had the highest Shannon index (1.142) due to the predominance of oak species in this forest. Because it is a shade-intolerant species (unlike beech, which is shade-tolerant), it has more species diversity than the forests in which beech is predominant.
Shannon index in the Nav forest was 0.439, which is the lowest value after Haftkhal forest. Despite this site being in the west of Gilan province, and the expectation of a high index of diversity, and in line with the general trend of biodiversity from west to east in the Hyrcanian forest, this decrease can be considered to result from intense exploitation of these forests. Also, the reason for the sharp decline in the diversity index in the Haftkhal forest site can be considered a consequence of severe exploitation.
Shannon index was 0.784 on the Chafroud site, located in the west of the Hyrcanian forests and in the middle highlands. The index was relatively high at this site.
In Mazandaran province, from west to east, respectively, this index for Ramsar, Sardabroud, Kheyroud and Haftkhal forest sites was 0.846, 0.681, 0.680 and 0.147, respectively. The trend of diversity from west to east was consistent, with the exception of the Sardabroud site, which, being at the mid elevation level, showed high biodiversity index. On the other hand, as it was said, HaftKhal forest site in Neka city had the lowest Shannon index, which apparently is related to both natural causes and management factors.
Extreme degradation and exploitation in the Neka region has led to high dominance for species such as beech, resulting in high dominance index and very low uniformity. In most of the sample plots, one or two dominant species with high frequency (beech species) were present and the presence of other species was low or not observed at all, which caused a decrease in uniformity and increased dominance in this area. In addition, intense management and exploitation play an important role in relation to the low diversity index in the Neka forest area, and another important factor in this regard is environmental influences, including elevation. This site is located at a relatively high elevation as compared to the other two areas. This, as well as the dominance of the northern direction in this region, can be another factor for the dominance of beech species and overcoming competition with other species.
Comment: Together with the criticism I referred above to the models, I think that the low quality of this section affects the suitability of your paper. I see also that very few references are reported in this section. I suggest you re-writing this section discussing your results in light of literature researches.
Response: revised as suggested and we re-wrote this section:
The reason for the superiority of the RF model over other models can be that this model is more powerful than other machine learning methods in the reducing data and that it is also less sensitive to parameter adjustment and is able to evaluate the importance of variables. In addition, it generates an internal unbiased estimate of the generalization error as the model building progresses [35, 36]. For example, Lee et al. [75] also used support vector regression (SVR), modified regression trees (RT) and random forest (RF) (as in the present study) in determining forest stand height using plot-based observations and airborne LiDAR data. Similar to the present study, which considers the efficiency of SVM and RF models in modeling forest biodiversity, they concluded that there was no statistically significant difference in plot height estimation between this models and all of them are acceptable. Also, in a study in Western Himalaya [76] that used machine learning including Classification Regression Tree (CART), Random Forest (RF), and Support Vector Machine (SVM) algorithms showed similar to the present study, according to the model accuracy statistics, and the authors concluded that, RF model, has a higher accuracy in Forest fire burn area. A study in Tasmania, Australia [77] used support vector regression (SVR), artificial neural networks (ANN), random forest (RF), and gradient boosted regression trees (GBRT) for mapping forest cover and exploring influential factors and their findings were in line with the results of the our study. They concluded in both fitting and projection accuracy, and required less computational costs, RF far outperformed the other three models.
4.3. Effect of biotic and abiotic factors on biodiversity index
The distinguishing feature of this study is the simultaneous study of the most important biotic and abiotic factors on biodiversity, which in comparison with related studies is one a few of studies that simultaneously considers important biotic and abiotic factors in biodiversity. Physiography affects the biodiversity of the forest. According to the results obtained from the models, especially the Random Forest model, which had better results than the other models, the effect of elevation was one of the most important factors affecting the biodiversity of forests in northern Iran. Changes of in elevation, the microclimatic, ecological and environmental conditions of the forest habitat, the structural condition of the massif changes in proportion to the local conditions [42]. With the change of the conditions governing the habitat and the land, the tree diversity changes and the amount of diversity increases in the favorable ecological and structural conditions of the land, and their amount decreases in unfavorable conditions In the Haft Khal region, due to being located at a higher altitude, there is less diversity in the region. In addition, studies such as [78] and [79] have stated that altitude is one of the most important factors affecting the biodiversity of the region, which is in line with the results of this study. Then biometric factors such as Basal area, Basal area of the thickest trees and tree diameters, physiographic factors such as slope and aspect and finally the type of species were mentioned as factors affecting biodiversity in the region. Species such as Fagus orientalis and Carpinus betulus are important and effective species in the region, while other species also had a positive effect on the modeling process [80]. Similarly patterns of diversity across different climate change of forest in China [23] concluded that both biotic and abiotic factors change the diversity of the forest community between different climatic zones. Also [20] and[21] considered both biotic and abiotic factors. Investigate the relative importance of mean air temperature, nitrogen availability and direct plant interactions in determining the millennial-scale population dynamics for four temperate tree taxa in the Scottish Highland concluded that all of factors are important. Also, as the sensitivity analysis in Figure 8 shows, elevation was the most important factor affecting biodiversity in the selected model (RF), the next factors were BA and BAL, respectively .BAL and Tree species were also important factor for the Random Forest model. Physiographic factors have a significant impact on the biodiversity index in forests, especially the elevation factor. Similar studies have shown that Physiographic factors play an important role as indicators of richness and diversity of species [24] determined affecting biotic and abiotic factors. In their study wind, topographic wetness index (TWI) and elevation were most important affecting factors in tree species richness variations. As seen in the results of this study, elevation was the main influential factor in RF models. This variable is a significant predictor affecting species diversity, as observed in previous studies [81–82]. Similarly, Aspect, slope and altitudinal variation in Ethiopian landscapes have influenced the existence of varied vegetation types and diversity [83]. The difference is that we used modeling to examine the relationship between environmental factors and the biodiversity index while they sampled quadrats and recorded data on species identity, abundance, elevation, slope and aspect. Also, they used different diversity indices and ordination techniques to analyze the data. Furthermore, Zhang et al. concluded elevation was one of the most important factors influencing community distribution and species diversity in the Balhus Mountains Reserve of Beijing, China, [6] the difference between their study and the present study was that they examined the functional diversity in the elevation gradient, while we used modeling using ML.
Comment: Just a minor observation: there is a mistake in the title: “…uneven aged and mixed forests…” not “…uneven egad and mixed forests…”
Response: We revised the manuscript as suggested.

Reviewer 2 Report
In this paper, the authors aimed to investigate the effect of biotic and abiotic factors on tree diversity of Hyrcanian forests in northern Iran. A combination of Machine Learning Methods was used for modelling and investigating the relationship between tree diversity and biotic and abiotic factors.
First of all, the article is well written and well organized. I think this article has good potential but, before being considered ready, some aspects need to be clarified and improved.
1)In my opinion, the paper title is too long. An appropriate title should not exceed ten words, be clear, informative and reflect the work. A suggested title would be "Assessing biotic and abiotic effects on Biodiversity index using Machine Learning".
2)The paper has many typos and grammatical mistakes. For example, in the abstract line 21 15,988 trees was measured=> 15,988 trees were measured. In line 22 A combination of Machine Learning Methods were used=> A combination of Machine Learning Methods was used. In line 26 diameter at breast height (DBH) in plot=>diameter at breast height (DBH) in the plot. In lines 30 and 31. Based on results of the RF method, elevation and BAL were recognized as the most influential factors defining variation of tree diversity=>Based on the results of the RF method, elevation and BAL were recognized as the most influential factors defining the variation of tree diversity.
3)I suggest you expand the literature review in the introduction to highlight the impact of the work.
4)Please add a separate section entitled, "Related Work", to transit smoothly in the following parts of the work.
5)The authors need to elaborate, why the specific ML models are selected. Moreover, the authors are requested to represent the cognitive features used to build different ML models in the current study.
6)For discussion, I would like to know the limitations and the potential issues of this study. Furthermore, the discussion part needs to be more technical and also highlight future works.
7)The conclusion can be further improved to elaborate on and highlight the results found in the simulation models. Moreover, conclusions can discuss future research directions and extensions of the study.
To sum up there is a clear presentation of the results and their commentary. Despite the technical contribution is limited, the authors made a solid description of the studying problem.
Author Response
Dear Dr Editor in chief and Dear Reviewers
Thank you for considering a revised version of our previous paper " Assessing biotic and abiotic effects on tree species richness in eight uneven aged and mixed forests using a machine learning approach” (Manuscript ID: forests-1159464) that incorporates the many useful comments and suggestions of the reviewers. Detailed corrections are listed below point by point. Based on the reviewer comments, we have substantially modified the paper.
Yours sincerely,
Reviewer #1:
In my opinion, the paper title is too long. An appropriate title should not exceed ten words, be clear, informative and reflect the work. A suggested title would be "Assessing biotic and abiotic effects on Biodiversity index using Machine Learning".
Response: We revised the manuscript as suggested.
Assessing biotic and abiotic effects on Biodiversity index using Machine Learning
The paper has many typos and grammatical mistakes. For example, in the abstract line 21 15,988 trees was measured=> 15,988 trees were measured. In line 22 A combination of Machine Learning Methods were used=> A combination of Machine Learning Methods was used. In line 26 diameter at breast height (DBH) in plot=>diameter at breast height (DBH) in the plot. In lines 30 and 31. Based on results of the RF method, elevation and BAL were recognized as the most influential factors defining variation of tree diversity=>Based on the results of the RF method, elevation and BAL were recognized as the most influential factors defining the variation of tree diversity.
Response: thank you, We revised the manuscript as suggested and corrected grammatical mistakes.
I suggest you expand the literature review in the introduction to highlight the impact of the work.
Response: We revised the manuscript as suggested. We added several References
According to related researches, studies that have considered both biotic and abiotic factors on biodiversity at the same time are limited, and this is one of the unique aspects of this study.
Please add a separate section entitled, "Related Work", to transit smoothly in the following parts of the work.
Response: We revised the” introduction” and added related work as sugessted:
Many studies have considered abiotic factors of forest by numerical methods and related them to tree growth measured in the field. [10-14], Also, various studies have considered only biotic factors and their effect on forest biodiversity such as [15, 16]. Therefore, abiotic environmental effects on species distributions and their ability to sustain viable populations in specific environmental configurations are not evaluated [17]. However, the species diversity in a given site is also influenced by other species through all interactions such as trophic, non-trophic and competitive [18, 19].
There are few studies that have considered the impact of biotic and abiotic factors on biodiversity that change simultaneously, such as [20] and [21]. Some studies indicate that biodiversity of a forest is affected by many factors, such as local climatic conditions, soil characteristics, biodiversity, and even the type of management practices employed [22]. Environmental factors are key variables that can help determine the diversity and distribution of plant species. For example: an analysis of patterns of diversity across different climate conditions of forest in China, which used PCA analysis to build the compound habitat gradient and biotic and abiotic factors, concluded that both biotic and abiotic factors influence the diversity of the forest community across different climatic zones [23].
Research studies that have considered both biotic and abiotic factors on biodiversity at the same time are limited, and this is one of the unique aspects of this study.
The authors need to elaborate, why the specific ML models are selected. Moreover, the authors are requested to represent the cognitive features used to build different ML models in the current study.
Response: We revised the manuscript as suggested
ML models have many advantages that justify their choice for modeling forest features such as biodiversity. For example, the Random Forest approach is less sensitive to parameter adjustments than other ML models and it provides an assessment of the importance of variables. This model, is more powerful in reducing data than other models, and it is more accurate than decision trees; it also generates an internal unbiased estimate of the generalization error as the model building progresses [35, 36].
SVM model another ML model that we used in this research. SVM models can be applied for solving nonlinear, regression and density estimation problems, and they are very useful in forest modeling. In addition, they use kernel functions in the form of points to project the multidimensional space of data, and then find the best classification of the hyperplane [37].
The K-nearest neighbors (KNN) model was also applied in this study. KNN is one of the easiest and simplest ML models and this is one of the advantages of using it. In this model, no hypothesis about the distribution of prediction variables is required and it can be applied to both single and multivariate predictions [38].
Another method, the GAM model, was also considered as it can limit the error in prediction of a dependent variable Y by assessing unspecific functions which are connected by means of a link function with the dependent variable. Providing a flexible specification of response by defining the model in terms of a smooth function is an additional advantage of the GAM model. [39]
For discussion, I would like to know the limitations and the potential issues of this study. Furthermore, the discussion part needs to be more technical and also highlight future works.
Response: We re-write discussion section as suggested:
4.1. Trend of tree diversity in the Hyrcanian forest
The Hyrcanian forest provides a wide range of ecosystem services of great value, including landscape, climate and water regulation, supporting atmospheric quality, timber and non-timber forest products, opportunities for tourism, recreation, health and wellbeing and spiritual experiences. The diversity of tree species is the basis of the biodiversity of the whole forest, because trees provide resources and habitats for other forest species. Species diversity in the forest changes under the influence of various factors [42]. Biodiversity ensures flexibility and adaptability of forest ecosystems, which protects the environment and leads to sustainable forest management [68]. Currently, considering biodiversity in forest management, along with other accepted economic and environmental criteria in the world, it is believed that in order to achieve the goals of sustainable forest management, forestry activities must be in line with environmental issues, especially plant biodiversity [69]. Also, the use of biodiversity indicators to evaluate different functions that have been developed for ecological indicators allows study of environmental characteristics, forest management [70; 71), and conservation [72] as applied to ecosystems. Because plants are the result of the environmental characteristics of each region, they are a full-fledged mirror of the habitat characteristics of that region [73]. Therefore, the study of plant composition and plant biodiversity can be used as an appropriate guide in ecological evaluations and study of biodiversity in each region. According to the results of the present study, biodiversity from west (Gilan province) to east (Golestan province) often has an increasing trend in the Hyrcanian forest, this is an important factor in creating this trend. In addition, temperature increases from west to east as does biodiversity [23]. But the annual precipitation decreases from 1345 mm at the Asalem forest in Gilan province at the westernmost point to 524 mm at the Loveh forest in Golestan province in the easternmost point. In addition, there are other factors that increase or decrease biodiversity in a forest site, the most important of which is elevation, Species diversity decreases with increasing elevation. But there are exceptions to this trend seen in this study; Loveh forest, despite being located in the easternmost point of the study, had the highest Shannon index (1.142) due to the predominance of oak species in this forest. Because it is a shade-intolerant species (unlike beech, which is shade-tolerant), it has more species diversity than the forests in which beech is predominant.
Shannon index in the Nav forest was 0.439, which is the lowest value after Haftkhal forest. Despite this site being in the west of Gilan province, and the expectation of a high index of diversity, and in line with the general trend of biodiversity from west to east in the Hyrcanian forest, this decrease can be considered to result from intense exploitation of these forests. Also, the reason for the sharp decline in the diversity index in the Haftkhal forest site can be considered a consequence of severe exploitation.
Shannon index was 0.784 on the Chafroud site, located in the west of the Hyrcanian forests and in the middle highlands. The index was relatively high at this site.
In Mazandaran province, from west to east, respectively, this index for Ramsar, Sardabroud, Kheyroud and Haftkhal forest sites was 0.846, 0.681, 0.680 and 0.147, respectively. The trend of diversity from west to east was consistent, with the exception of the Sardabroud site, which, being at the mid elevation level, showed high biodiversity index. On the other hand, as it was said, HaftKhal forest site in Neka city had the lowest Shannon index, which apparently is related to both natural causes and management factors.
Extreme degradation and exploitation in the Neka region has led to high dominance for species such as beech, resulting in high dominance index and very low uniformity. In most of the sample plots, one or two dominant species with high frequency (beech species) were present and the presence of other species was low or not observed at all, which caused a decrease in uniformity and increased dominance in this area. In addition, intense management and exploitation play an important role in relation to the low diversity index in the Neka forest area, and another important factor in this regard is environmental influences, including elevation. This site is located at a relatively high elevation as compared to the other two areas. This, as well as the dominance of the northern direction in this region, can be another factor for the dominance of beech species and overcoming competition with other species.
4.2. Machine learning approach to modeling diversity
In this research, four ML models were implemented including Generalized Additive Model (GAMs), Support Vector Machine (SVM), Random Forest (RF) and K-Nearest Neighbor Algorithm (KNN). Among ML models, RF model with R2 0.59 and RMSE 0.28 was the best model followed by SVM, Nearest Neighbor Algorithm, and GAM models with R2 0.41, 0.30 and 0.17, respectively. Thus, as can be seen, the RF model was the best model among the ML models, which was similar to the results of many researchers in this field [74,25]. The reason for the superiority of the RF model over other models can be that this model is more powerful than other machine learning methods in the reducing data and that it is also less sensitive to parameter adjustment and is able to evaluate the importance of variables. In addition, it generates an internal unbiased estimate of the generalization error as the model building progresses [35, 36].For example, Lee et al. [75] also used support vector regression (SVR), modified regression trees (RT) and random forest (RF) (as in the present study) in determining forest stand height using plot-based observations and airborne LiDAR data. Similar to the present study, which considers the efficiency of SVM and RF models in modeling forest biodiversity, they concluded that there was no statistically significant difference in plot height estimation between this models and all of them are acceptable. Also, in a study in Western Himalaya [76] that used machine learning including Classification Regression Tree (CART), Random Forest (RF), and Support Vector Machine (SVM) algorithms showed similar to the present study, according to the model accuracy statistics, and the authors concluded that, RF model, has a higher accuracy in Forest fire burn area. A study in Tasmania, Australia [77] used support vector regression (SVR), artificial neural networks (ANN), random forest (RF), and gradient boosted regression trees (GBRT) for mapping forest cover and exploring influential factors and their findings were in line with the results of the our study. They concluded in both fitting and projection accuracy, and required less computational costs, RF far outperformed the other three models.
4.3. Effect of biotic and abiotic factors on biodiversity index
The distinguishing feature of this study is the simultaneous study of the most important biotic and abiotic factors on biodiversity, which in comparison with related studies is one a few of studies that simultaneously considers important biotic and abiotic factors in biodiversity. Physiography affects the biodiversity of the forest. According to the results obtained from the models, especially the Random Forest model, which had better results than the other models, the effect of elevation was one of the most important factors affecting the biodiversity of forests in northern Iran. Changes of in elevation, the microclimatic, ecological and environmental conditions of the forest habitat, the structural condition of the massif changes in proportion to the local conditions [42]. With the change of the conditions governing the habitat and the land, the tree diversity changes and the amount of diversity increases in the favorable ecological and structural conditions of the land, and their amount decreases in unfavorable conditions In the Haft Khal region, due to being located at a higher altitude, there is less diversity in the region. In addition, studies such as [78] and [79] have stated that altitude is one of the most important factors affecting the biodiversity of the region, which is in line with the results of this study. Then biometric factors such as Basal area, Basal area of the thickest trees and tree diameters, physiographic factors such as slope and aspect and finally the type of species were mentioned as factors affecting biodiversity in the region. Species such as Fagus orientalis and Carpinus betulus are important and effective species in the region, while other species also had a positive effect on the modeling process [80]. Similarly patterns of diversity across different climate change of forest in China [23] concluded that both biotic and abiotic factors change the diversity of the forest community between different climatic zones. Also [20] and[21] considered both biotic and abiotic factors. Investigate the relative importance of mean air temperature, nitrogen availability and direct plant interactions in determining the millennial-scale population dynamics for four temperate tree taxa in the Scottish Highland concluded that all of factors are important. Also, as the sensitivity analysis in Figure 8 shows, elevation was the most important factor affecting biodiversity in the selected model (RF), the next factors were BA and BAL, respectively .BAL and Tree species were also important factor for the Random Forest model. Physiographic factors have a significant impact on the biodiversity index in forests, especially the elevation factor. Similar studies have shown that Physiographic factors play an important role as indicators of richness and diversity of species [24] determined affecting biotic and abiotic factors. In their study wind, topographic wetness index (TWI) and elevation were most important affecting factors in tree species richness variations. As seen in the results of this study, elevation was the main influential factor in RF models. This variable is a significant predictor affecting species diversity, as observed in previous studies [81–82]. Similarly, Aspect, slope and altitudinal variation in Ethiopian landscapes have influenced the existence of varied vegetation types and diversity [83]. The difference is that we used modeling to examine the relationship between environmental factors and the biodiversity index while they sampled quadrats and recorded data on species identity, abundance, elevation, slope and aspect. Also, they used different diversity indices and ordination techniques to analyze the data. Furthermore, Zhang et al. concluded elevation was one of the most important factors influencing community distribution and species diversity in the Balhus Mountains Reserve of Beijing, China, [6] the difference between their study and the present study was that they examined the functional diversity in the elevation gradient, while we used modeling using ML.
The results of various studies show that the middle elevations generally have the highest index of richness and species diversity, which in the present study is the reason for the high level of these indicators in Sardabroud forest site.
In Haftkhal in Neka forest site, elevation is relatively higher than other areas; this, as well as the dominance of the northern direction in this region, can be another factor for the dominance of the beech species and its dominance in competition with other species. In fact, less uniformity at high elevations is due to the abundance and dominance of beech species.
In general, the results of similar studies show that lower temperatures and slower melting of ice in these areas, especially at high elevations, causes less variety in diversity. These slopes are more humidity and colder causing the dominance of beech species and as a result reduce the uniformity index. Although, it is worth mentioning that in different regions, due to their climatic and geological characteristics and geographical location, different results are obtained about aspect, but usually diversity is greater in aspects with higher humidity and temperature.
Another important factor that affects the diversity index, along with natural factors, is management and conservation practices in the forests. In fact, the huge difference in biodiversity index between forest sites with similar elevation and environmental conditions, despite the relative similarity of climatic, physiographic and biological conditions, can be attributed to the management style in these forests.
Therefore, in general, the Shannon diversity index in the managed and protected forest area is significantly larger compared to other areas, which indicates that tree felling and human pressure in the area has resulted in more heterogeneity in the number and diversity of reproduction of different species.
It is necessary to mention that it seems that other environmental factors such as humidity, precipitation, temperature and soil in the studied forest sites, one would expect that the modeling coefficient will increase. Thus, if a model is developed that includes all the other abiotic factors (mentioned), the R2 should be much higher. In our research, results show the capability of some machine learning techniques to produce accurate estimates of biodiversity index in forest sites and to identify important variables (e.g., elevation, BAL). Although it cannot be said that RF techniques may always be better than other machine learning methods, our results showed a high coefficient of determination and low RMSE than other ML methods evaluated. RF models incorporate information of environmental variables into models and extract information directly from data without any pre-defined assumptions of the phenomenon being investigated to significantly improve the quality of the models [13,72]. the same results were achieved using ML and geo statistical methods [34] to predict aboveground biomass in Chinese forests, which concluded that the Random Forest created a reliable and accurate method for AGB mapping in subtropical forest regions with complex topography.
The conclusion can be further improved to elaborate on and highlight the results found in the simulation models. Moreover, conclusions can discuss future research directions and extensions of the study.
Response: We re-write conclusion section as suggested:
The main goal of natural resource management is to preserve biodiversity in natural ecosystems, so that habitats with more biodiversity have more ecological stability and fertility than areas with less biodiversity, and more stable ecosystems will be more dynamic. In this study, by combining biotic and abiotic factors in ML models, we analyzed their relationship to the biodiversity index across 8 forest sites in the Hyrcanian forest in northern Iran. Four machine learning algorithms including GAM, SVM, RF, and Nearest Neighbor Algorithm were used to model tree diversity. The results showed that machine learning methods, especially the Random Forest and Support Vector Machine, were more accurate than other methods. Based on results of RF model, elevation, BA, and BAL, were recognized as the most influential factors defining variation of tree diversity in the Hyrcanian forests. Also, in this study, we simultaneously examined the important biotic and abiotic factors in relation to the biodiversity index, which distinguishes this study from similar studies.
Machine learning techniques can often superior to traditional methods when assumptions model for applying parametric procedures are not validated Also, flexibility, accuracy, and the ability to model complex and nonlinear relationships are features of ML methods.

Round 2
Reviewer 1 Report
Dear authors,
in general, I think that this new version of the manuscript is improved, especially in the discussion section. You accepted my suggestions and included the main tree species among the biotic variables in the models. The adoption of sub-chapters in the discussion gives a better focus on the various aspects addressed in the paper, with several references to the literature.
However, I think that the manuscript should be strongly revised in its English language. In many sentences, I found it very difficult to fully understand what you would like to state.
In addition, I report below several minor remarks:
Line 54: remove “such as” and keep only the references.
Line 60: remove “such as” and “and” and keep only the references.
Lines 60-62: you cite only a reference. Instead of “some studies” specify: “Pilli and Pase (2018) indicate..”
Table 1: add also the quantitative variable “number of trees per hectare “and the qualitative variable “Tree Species”.
Lines 524-525: remove “such as”. You can write: “In addition, some studies [78,79] have stated…”
Lines 533-534: specify in detail who and where. Cite only references is reductive.
Line 534-537: add the citation.
Line 552: add the year of the reference.
Author Response
Dear Dr Editor in chief and Dear Reviewer
Thank you for considering a revised version of our previous paper " Assessing biotic and abiotic effects on tree species richness in eight uneven aged and mixed forests using a machine learning approach” (Manuscript ID: forests-1159464) that incorporates the many useful comments and suggestions of the reviewers. Detailed corrections are listed below point by point. Based on the reviewer comments, we have modified the paper.
Yours sincerely,
in general, I think that this new version of the manuscript is improved, especially in the discussion section. You accepted my suggestions and included the main tree species among the biotic variables in the models. The adoption of sub-chapters in the discussion gives a better focus on the various aspects addressed in the paper, with several references to the literature.
However, I think that the manuscript should be strongly revised in its English language. In many sentences, I found it very difficult to fully understand what you would like to state.
Thank you very much for your great comment, We revised the manuscript as suggested several times.
In addition, I report below several minor remarks:
Line 54: remove “such as” and keep only the references
We revised the manuscript as suggested..
Line 60: remove “such as” and “and” and keep only the references.
We revised the manuscript as suggested.
Lines 60-62: you cite only a reference. Instead of “some studies” specify: “Pilli and Pase (2018) indicate..”
We revised the manuscript as suggested.
Table 1: add also the quantitative variable “number of trees per hectare “and the qualitative variable “Tree Species”.
We revised the manuscript as suggested. We added number of trees per hectare in table 1.
Lines 524-525: remove “such as”. You can write: “In addition, some studies [78,79] have stated…”
Thank you,we revised the manuscript as suggested.
Lines 533-534: specify in detail who and where. Cite only references is reductive.
We revised the manuscript as suggested.
Line 534-537: add the citation.
We revised the manuscript as suggested.
Line 552: add the year of the reference.
We revised the manuscript as suggested.
